# Students' Perceived M-Learning Quality: An Evaluation and Directions to Improve the Quality for H-Learning

**Syed Faizan Hussain Zaidi** [1,*] , **Atik Kulakli** [1] , **Valmira Osmanaj** [1] **and Syed Ahasan Hussain Zaidi** [2]

1   College of Business Administration, American University of the Middle East, Egaila 54200, Kuwait;
    atik.kulakli@aum.edu.kw (A.K.); valmira.osmanaj@aum.edu.kw (V.O.)
2   Department of Applied Sciences, RVIT College (Dr. A. P. J. Abdul Kalam Technical University),
    Lucknow 695016, India; sahzaidi123@gmail.com
*   Correspondence: syed.zaidi@aum.edu.kw

**Abstract:** The COVID-19 pandemic has transformed the paradigm of the higher education sector and has instigated a speedy consumption of a diverse range of mobile learning software systems. Many universities were adhering to online modes of education during the pandemic; however, some of the universities are now following hybrid modes of learning, termed h-learning. Higher education students spent two years of taking their classes online during the COVID-19 pandemic and have experienced various challenges. Simultaneously, the main challenge for higher education institutions remains how to consistently offer the best quality of students' perceived m-learning and maintain continuance for the new shift towards hybrid learning. Hence, it becomes essential to determine the m-learning quality factors that would contribute to maintaining superior m-learning quality in higher education during the COVID-19 pandemic and afterwards via a hybrid mode of learning. Thus, the m-learning quality (MLQual) framework was conceptualized through an extensive review of the literature, and by employing survey-based quantitative research methods, MLQual was validated via structural equation modeling (SEM) techniques. The outcome of this research yielded the MLQual framework used to evaluate the students' perceived m-learning quality and will offer higher education practitioners the chance to upgrade their higher education policies for h-learning accordingly. With the preceding discussion, it is evident that evaluation of the students' perceived m-learning quality factors in higher education is always a question that should be researched adequately. Determination of such m-learning quality factors is essential in order to offer significant directions to the higher education practitioners for improving both the quality and delivery of m-learning and h-learning. Consequently, the present study embraces two key objectives: First, to identify and evaluate the m-learning quality factors which could be employed to improve the quality of m-learning. Second, to propose the MLQual framework for the evaluation of students' perceived m-learning quality.

**Keywords:** m-learning; e-learning; h-learning; educational technology; higher education; evaluation; m-learning quality; system quality; service quality

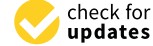



## 1. Introduction

The COVID-19 pandemic has transformed the paradigm of the academic sector and caused the delivery of traditional on-campus education to shift towards mobile learning (m-learning) practices [1,2]. M-learning, specifically within various university settings, has developed as a prevalent tool and unconventional learning method, and due to its effective multi-functional characteristics, it has been assisting students to bridge their educational and learning gap throughout the COVID-19 pandemic [1,3]. Various higher education institutions have experimented with online exams prior to the COVID-19 pandemic [4] and have utilized various proctoring tools, making online exams more efficient compared with the traditional paper-based examination approach [5]. Many academics furnished compelling justifications related to the advantages of m-learning encompassing mobility,

user's independence, self-learning, improvement in student–teacher communication, online engagement, and rapid exchange of information [6]. Mobile technologies in today's digital diaspora offer innovative digital capabilities including a broader color display, high screen resolution, extended memory size, multimedia enabled interfaces, WiFi, and rapid browsing capabilities in a range of mobile devices [7,8]. Conventional styles of learning have been restructured and reinvented due to smart mobile devices letting the students become involved in borderless uninterrupted learning practices [9].

According to Behera [10], m-learning is regarded as an addition to e-learning, and the quality of m-learning can be improved by being cognizant of the unique constraints and advantages of mobile devices. Basak et al. [11] stated in their research that "the m-learning is the subset of e-learning, which includes the m-learning and the online environments as well". M-learning features enhance interactions among remotely located students groups to carry out group work through resource sharing. Furthermore, m-learning also develops students' collaborative learning capacities and creativity [12,13]. M-learning combines blended/hybrid, interactive, ubiquitous, personal, instantaneous information retrieval, and collaborative aspects that allow students to develop their own learning space; hence, actual learning experience enriches the students' enjoyment [14]. On the other side, h-learning is now the choice of many universities in some countries where substantially decreased rates in COVID-19 cases were noticed [15]. Siegelman [16] articulates that h-learning and blended learning approaches are substitutable. Several universities adopted online teaching as a result of the impact of the COVID-19 pandemic and continued with m-learning options, whereas other universities opted for h-learning options via combining on-campus and online sessions.

Universities should understand and devise ways to effectively implement m-learning services and exploit its possible features to facilitate students. Additionally, universities should examine students' perspectives towards m-learning systems and service quality [17]. The majority of research studies are looking into the students' acceptance or adoption of m-learning, and have used various forms of technology acceptance models (TAMs) [8,18]. Despite the growing use and popularity of m-learning since the COVID-19 pandemic, the research gap related to students' perspectives, students' m-learning styles, and students' attainment remains unexplored [9]. Sahu [19] highlighted the significance of quality in online education during the COVID-19 pandemic and addressed the requirement to pay appropriate attention towards maintaining quality while delivering online sessions. He also emphasized considering the quality-related issues, which were raised while making a shift towards online modes of teaching. The application of m-learning in higher education settings is drawing academic researchers' attention to the exploration of m-learning quality, effectiveness, and acceptability in educational settings [2,20]. According to Laksana [20], regardless of the benefits of m-learning, if the students are reluctant to fully involve themselves in online studies and do not show active participation, then prospective benefits of m-learning cannot be apprehended. He further mentions that "students' perceptions of quality of online academic interactions are only 69%".

The abovementioned arguments clearly indicate that students not only need mobile devices, comprising smart phones, tables, and laptops during m-learning, but they also need m-learning systems of the utmost quality. Universities' administrative authorities and m-learning system designers have to comprehend students' perceptions towards m-learning and related quality factors which could provide superior m-learning quality to the students during the COVID-19 pandemic, and thereafter in the h-learning mode. Thus, the proposed research contributes to the m-learning literature through accomplishing two objectives:

(1) To identify and evaluate the m-learning quality factors which could be employed to improve the quality of m-learning in h-learning mode.
(2) To propose an MLQual framework for the evaluation of students' perceived m-learning quality.

M-learning is more than just a new method of passively accessing educational resources and mobile devices for m-learning work as intermediary between students and instructors. Within a flexible time and location-constrained setting, instructors and students should successfully complete their teaching and learning objectives. At the same time, the online educational content offered, students' motives, learning attitudes, and conventional learning habits build the students' interest in m-learning. Consequently, students' perceptions about m-learning quality is a question to be explored carefully. As discussed previously, regardless of the intensive usage of m-learning during the COVID-19 pandemic, research on the students' behavior and perceptions on the m-learning quality is still unexplored. Thus, the goal of the study is to understand the students' perceptions towards m-learning quality in higher education perspectives and how to improve the quality of educational delivery in the modes of m-learning and h-learning. M-learning system designers have to realize students' perceptions towards m-learning and related m-learning quality factors which could provide superior m-learning quality to the students' m-learning and hybrid modes of learning. As a result, this study identified the following research questions:

(1) What are the factors comprising the students' perceptions towards m-learning quality within the context of higher education?

(2) Are there significant associations between the m-learning quality factors and students' utilization of the m-learning system?

This study proposes a novel framework, MLQual, for evaluating the quality of m-learning and recommends guidelines to improve h-learning. This paper is organized into a number of sections. The literature review section discusses the learning techniques in higher education, h-learning, and m-learning quality evaluation models. After a thorough review of the literature, the framework development section conceptualizes and proposes the MLQual framework and develops various hypotheses. The methodology section discusses the quantitative research method which is utilized in present research. The data analysis and results discussion sections confirm the validation of the proposed MLQual framework. Finally, implications, limitations, future research directions, and the conclusion are presented in succeeding sections.

## 2. Related Studies

The first section of the literature highlights various learning techniques in higher education and from the viewpoint of our present study. The second section highlights m-learning quality models and frameworks reported in higher education.

### 2.1. Learning Techniques in Higher Education

The effect of technology is inevitable in the transition of education from the traditional to the online or mobile modes of learning. Today's online learning environment has technologically developed, and it supplements traditional learning by offering a broad range of online resources to a large audience [5,21]. Al-Hunaiyyan et al. [22] emphasized the significance of switching from traditional learning tactics to m-learning strategies, which provide students with flexible learning environments and methodologies that expose them to new learning experiences. While taking the shift from traditional to m-learning or hybrid learning environments, the educational sector is now embracing cutting-edge technology including collaborative video conferencing platforms such as "Google Classroom, Zoom, Moodle Video, Microsoft Team, and Webex" [17,23]. Consequently, this is enhancing the learning pattern globally and being applied in various educational disciplines to enhance the students learning. Preceding arguments clearly explain that such a vital addition to traditional learning transforms traditional learning practices to online and hybrid modes of learning.

There has been notable expansion in both education and information technology, with the current decade seeing the most significant increase in and the emergence of m-learning [24]. M-learning is an essential medium in the fast-expanding electronic learning

sector. A "theory of m-learning" suggests the substantial learning of students which takes place in classrooms and outside of the classroom, which means in online environments [25]. As mobile technology develops, m-learning uses wireless technologies and portable devices to educate students regardless of time and location. Using the distinct advantages of mobile devices, m-learning is a novel approach to the education system. The productive acceptance of m-learning systems depends on student's perceptions [26]. In order to encourage the implementation of m-learning features, previous studies have attempted to characterize it in the form of theories and frameworks.

Bashir et al. [27] explain the h-learning model or "hyflex model" which combines face-to-face course delivery with online students' presence simultaneously, so all the enrolled students of the class appear in both modes simultaneously. Other studies also support that h-learning is the form of face-to-face, and online learning combinations, also called blended learning [15,28,29]. With the growth and development of blended learning research, more and more voices are joining the discourse. Butz and Stupnisky [30] used a multiphase mixed method to investigate the linkages between students' need fulfillment, motivation, and accomplishment in synchronous h-learning settings using Ryan and Deci's [31] "Self-Determination Theory" to find out on-campus teaching and online learning with the Web conferencing platforms differentiated. The findings suggest four areas influencing h-learning: "peer relatedness, technology influence, instructor impact, and program structure". Porter et al. [32] proposed a framework for institutional blended learning adoption which identifies three stages: "awareness", "adoption", and "mature implementation". The authors applied their framework and Rogers' [33] "Diffusion of Innovations Theory" to form the organizational strategies to adopt the theory to facilitate blended learning.

Abu Saa et al. [34] studied the factors that affect students' performance in university education. The "predictive data mining" results showed that the most prevalent variables are divided into four major categories: students' primary grades and academic performance; students' participation in online learning; students' demographics; and students' social data. Similarly, Flores et al. [35] studied the effects of the COVID-19 pandemic on online teaching and learning. The article pulls from a larger body of research that examined how college students adjusted to the closure of their institution and how they viewed their involvement in online teaching and learning via the Internet. Findings indicated that contextual and individual factors contributed to students' "positive or negative adaptation" during the facility's closure. Moreover, the factors contributing to the results include "institutional and pedagogical" reactions, "individual self-regulatory" and "socioemotional competencies", and enough resources.

Dospinescu and Dospinescu [36] conducted a comparative study to explore the perception of e-learning tools, namely Moodle and Blackboard. The study took place in Romania and Moldova during the same period, and factors were evaluated accordingly for both countries regarding teaching and evaluation processes. The findings indicated a correlation between the determinants such as "automation of evaluation process, customized content, online tutor/mentor interaction, quality of administrative services, quality of information, and video format content". Moreover, in their respective study, Ouyang et al. [37] explored how artificial intelligence (AI) applications affect online higher education. Results indicated that the learning and satisfaction levels, recommendations of resources, automated assessment and evaluations, responses of feedback, and experience are standard functions and features. At the same time, traditional AI technologies are also adopted for high-quality services. Therefore, improved academic performance encourages "online participation and involvement".

## 2.2. Concept of H-Learning

To illustrate h-learning, numerous definitions and explanations have been developed. As a result, there are arguing explanations of what precisely constitutes h-learning. As a result, adopting a clear description is essential. As previously explained, h-learning

integrates both face-to-face and online learning. According to Hentea et al. [38], "the concept of Hybrid Learning or Blended Learning refers to the combination of an online learning environment by gaining the flexibility of distance or outside of classroom learning, and a face-to-face (F2F) classroom instruction". The terms "h-learning" and "blended learning" relate to the same techniques which combine online, and on-site teaching modalities and the terms used interchangeably geographically [39]. Qi and Tian [40] stressed four properties of h-learning including: a combination of group and learning, a combination of synchronous and asynchronous learning, a combination of group and self-paced learning, and a combination of formal and informal education that incorporates lifetime learning. When in-person and online instruction are used simultaneously, an instructor can instruct students both in-person and online at the same time. As a result, some students attend the lesson in person while others participate electronically using tools like conferencing systems. Students participate more actively in the learning process while using h-learning [41]. Irvine et al. [42] stressed the importance of good audio and good video quality. To improve students' learning and experience, academic tasks and activities carried out in an h-learning setting should encourage peer collaboration and increase student involvement. The abrupt switch to h-learning platforms has pushed the teaching staff to come up with innovative methods for delivering their courses [43].

Hybrid technique that combines the cognitive apprenticeship model with the collaborative learning strategy is proposed for performing web-based problem-solving activities is offered [44]. Similarly, integrative models are employed to promote hybrid learning (h-learning), a contextual, transformational, collaborative, and situated learning technique that can address engineering's growing complexity [45]. The advancement of technology enabled new opportunities in healthcare education, such as online, blended, and flipped models. In response to the COVID-19 epidemic, physical therapy institutions have used h-learning [46]. Nørgård [47] studied how to organize the form of h-learning and handle the new requirements according to pandemic conditions. Learning suppliers and organizations have to rethink and reorganize learning to be more online and "pandemic-friendly" in order to continue learning in a pandemic environment.

### 2.3. M-Learning Quality Models

M-learning quality is a vital aspect in the higher education setting and its accomplishment significantly depends on the efficient delivery of the online learning experience to the students in compliance with international quality standards and norms. Achieving and maintaining quality performance in today's higher education environment is considered crucial, particularly when students are using mobile devices for learning purposes [48]. Furthermore, the authors argue that learning process quality is affected not only by the physical features of the environment where it is provided, but also the broader environmental factors, such as economic conditions and legislation [49]. Similarly, Usak et al. [50] proclaimed that the COVID-19 pandemic has been distressing the students' routine study and educational quality. Despite this, quality-related obstacles are less researched during the COVID-19 pandemic [51].

The concept of quality is difficult to describe correctly, particularly in higher education systems since educational establishments have complete sovereignty in determining their vision and goals [52]. Service quality in higher education is the difference between what any student anticipates and what they receive in the service delivery process [53]. Aburub and Alnawas [54] expressed the quality within the context of educational systems as a "degree of excellence" that educational institutions should enhance regularly. Moreover, Althunibat et al. [49] used the concept of quality of m-learning applications as "a degree of excellence of learning content quality and learning service quality of the m-learning system" (p. 2). According to Del Río-Rama et al. [55], the argument on "how to define service quality in higher education is still going on and as an outcome, no consensus exists on the most effective technique to define and quantify service quality". This is attributed to the complex

nature of the services, arising from its heterogeneity, inseparability of production and consumption, perishability, and intangibility [56,57].

In previous studies, the focus was on exploring the importance of quality-related factors in the context of e-learning quality [49,58,59], learning management system quality [6], and m-learning quality [60]. The m-learning quality model proposed by Almaiah and Alismaiel [61] combined TAMs along with DeLone and McLean's (D&M) revised model to evaluate the factors that determine students' acceptance of m-learning systems. On the other hand, Al-Nassar [62] adopted the revised D&M model to design and recommend a service quality model for m-learning, considering the effect of information and system quality, as perceived by university students. The SERVQUAL model [63] was utilized by Sumi and Kabir [59] to assess the perceived quality and satisfaction of the university students with regard to e-learning initiatives, implemented in the time of COVID-19 pandemic.

In consonance with the current literature, a significant upsurge is observed in the number of empirical studies focusing on the factors that affects the intention, acceptance, adoption, design, deployment, and usage of m-learning at higher education institutions [21,48,61]. Nevertheless, the number of studies that address the users' perceptions of factors that affect the m-learning quality are very scattered [49].

## 3. Theoretical Framework and Hypotheses Development

The SERVQUAL model [63] is the consumer perceived service quality measurement instrument that consists of five dimensions (reliability, tangibles, assurance, empathy, and responsiveness) that measures consumer perception and expectancy of offered services. Extending consumers' service expectancy–rejection paradigm in higher education becomes important to evaluate students' perceived m-learning quality. Hence, below is the detailed description of SERVQUAL dimensions along with auxiliary dimensions which are conceptualized and comprised in this study for the progression of SERVQUAL in the field of m-learning quality evaluation. Within the perspective of m-learning quality evaluation, functionality, collaboration, security, usefulness, mobile learning system quality (MLSysQ), m-learning service quality (MLSerQ), and perceived m-learning quality are the additional dimensions identified from the contemporary m-learning literature and incorporated in conceptualized MLQual framework. Figure 1 shows the hypothesized model for the evaluation of students' perceived m-learning quality.

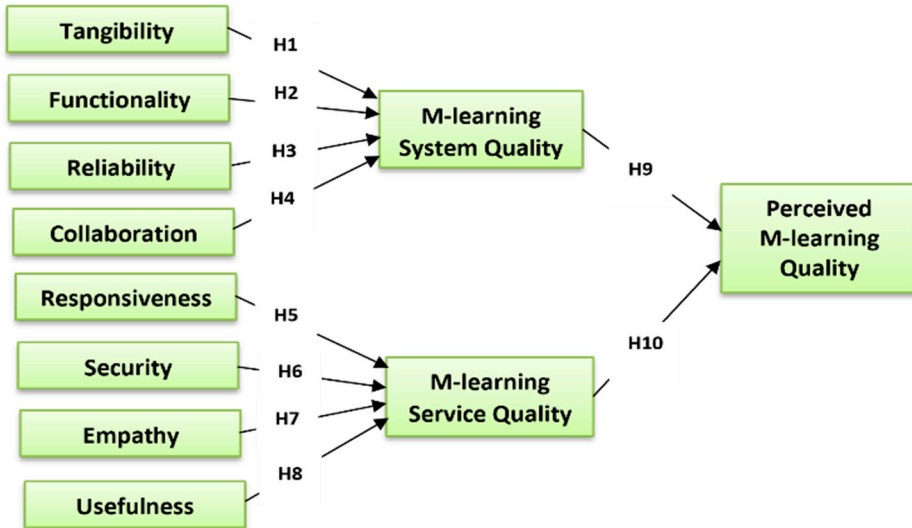

**Figure 1.** M-learning quality (MLQual) evaluation framework.

Hypothesized relationships among the proposed dimensions are illustrated below in detail.

### 3.1. Tangibility

Physical facilities, equipment, and appearance of personnel are the parts of SERVQUAL. Akhlaghi et al. [64] evaluated educational service quality and used tangibility as one of the dimensions; these were assessed using the appearance of physical facilities, equipment, personnel, and communication materials. Since the COVID-19 pandemic, students have been taking online classes and using mobile devices and software [26] which can be categorized as tangible objects for m-learning. As mobile devices, laptops, memory size, internet speed, and equipment are tangible objects, all of these constitute system quality during m-learning; hence, tangibility affects the system quality during online classes.

**H1.** *Tangibility as dimension affects the m-learning system quality during online classes in m-learning sessions.*

### 3.2. Functionality

Essential features of any system and their functionality play an imperative role in the m-learning process while using m-learning systems [11]. Educational establishments have been bound due to the COVID19 pandemic to improve site edification and adhered to offering online courses through the Web [26]. M-learning systems are web-based systems and can be accessed through Web applications or websites of each individual institution. Consequently, the presence of possible required Web functions for m-learning must be obtainable in m-learning Web-based software systems. Furthermore, m-learning software systems should offer user-friendly, interactive, flexible, and video-based learning environment while taking online classes. M-learning software systems should offer quick navigation back and forth amongst Web pages [65,66]. The preceding discussion clearly highlights the importance of functionality of an m-learning environment; henceforth, the functionality of an m-learning software system affects the quality of the m-learning system.

**H2.** *Functionality of m-learning software system affects the m-learning system quality in m-learning.*

### 3.3. Reliability

Salloum et al. [67] addressed the efforts of ensuring the reliability of e-learning systems for e-learning. Additionally, their study considered the difficulty in adaptation of the resources for e-learning systems. Henceforth, it becomes important to study reliability as a key dimension and its impact on m-learning system quality. Furthermore, Azeroual and Jha [68] consider correctness, completeness, consistency, and timeliness to be attributes of data reliability. Considering the preceding argument, the present study embraces the reliability of mobile devices and m-learning systems, which explains the ability to quickly upload and download homework, assignments, projects, and online exams without any inaccuracy and without the time and place restrictions students have in order to use m-learning systems.

**H3.** *Reliability of m-learning software system affects the m-learning system quality during online classes in m-learning sessions.*

### 3.4. Collaboration

Collaboration is an important aspect of m-learning which should allow a group of people to interact with each other; moreover, components such as collaboration, interactivity, and connectivity are considerably essential for effective m-learning [22]. This indicates that students and instructors should be able to collaborate to work on various activities, assignments, and projects. Even groups of students should be able to work in collaborative interactive working environments. M-learning software systems through their Web capabilities should be able to enrich students' engagement, interactivity, and collaborations, and these influence the quality of the m-learning software system. The following hypothesized relationship can be derived as mentioned below.

**H4.** *Collaboration in m-learning as significant dimension affects the m-learning software system quality in m-learning.*

*3.5. Responsiveness*

Responsiveness means willingness to assist consumers and offer timely services [68]. Therefore, within the context of m-learning, responsiveness is considered a significant factor as instructors and students are positioned at distant locations. Responsiveness in m-learning can be explained as attention promptly given to students towards their online queries. Responsiveness as a measure of m-learning service quality contributes to effective execution of the m-learning system [61]. The following hypothesized relationship can be derived as shown below.

**H5.** *The responsiveness factor in m-learning system affects the m-learning service quality in m-learning.*

*3.6. Security*

From the students' perspective security is defined as "the extent to which students believe that their online identity and transactions are secured, and privacy is retained" [58]. From the preceding statements it is clear that a student's personal information is always confidential and should not be available to others. Student's online transactions encompass the transactions of documents, files, and personal information, and security of theses have to be mentioned as part of students' engagement during m-learning. The institution's IT support should offer regular security updates of the m-learning software system. This should enhance the students' confidence and trust in m-learning. Hence, security is an essential facet in m-learning and considered to be a key determinant for assessing the quality of the m-learning service. The following hypothesized relationship can be derived as shown below.

**H6.** *Security in m-learning system is significant dimension affects the m-learning service quality.*

*3.7. Empathy*

The SERVQUAL model [63] includes empathy, defined as "the caring, individualized attention the firm provides to its customers", which means in m-learning context caring and individualized attention institutions should be provided to the students while conducting online classes [69]. For the purpose of evaluating the e-learning quality and students' satisfaction, Stodnick and Rogers [70] emphasized the importance of empathy and implicated it in their study as forecasting variable of e-learning quality. In the present study, m-learning quality assessment, empathy is included as an important predictor. Students' advising through m-learning software system, understanding students' needs, giving attention to individual attention online, and encouraging students to increase their participation in m-learning found to be significant measurement factors; hence, these are included when discussing empathy. The preceding discussion clearly reveals that empathy is a determining factor for m-learning service quality. The following hypothesized relationship can be derived as below.

**H7.** *Empathy from instructors in m-learning environment positively affects the m-learning service quality.*

*3.8. Usefulness*

When students take online courses, useful, up-to-date, complete, and relevant learning contents are then delivered by the instructors in order to improve the students' perceptions towards system utilization and service usefulness [71,72]. Sitar-Taut and Mican [73] stated that in blended learning, the appearance of the contents is equally essential in comparison to the contents of the course; this enhances the usefulness of the m-learning service among students. In the context of the present study, other than the usefulness of the relevant contents, pursuing weekly up-to-date tasks, complete information, time management,

speedy responses through m-learning software systems, and educational gap recovery during the pandemic are also important aspects of usefulness in m-learning service quality. The following hypothesized relationship can be derived as shown below.

**H8.** *Usefulness in m-learning as significant dimension affects the m-learning service quality.*

### 3.9. M-Learning System Quality

The system quality of the m-learning system is determined during the analysis, design, and implementation phases of the desired system; however, actual m-learning system quality is relatively established based on students' perceptions [6]. Sarrab, Hafedh and Bader [65] embraced "availability, usability, dependability, performance, and functionality" factors for evaluating the system quality of m-learning applications. Almaiah and Man [6] also assessed m-learning system quality using "functionality, accessibility, interactivity, interface design, and ease of use" factors from the students' perspective. In the present study, tangibility, functionality, reliability, and collaboration are considered as preceding dimensions, and the impact of these on the quality of the m-learning system is studied. The compatibility of the M-learning systems with android or IOS systems on mobile devices, technicality of m-learning application, screen resolution, and processing speed for carrying out heavy transactions are employed as factors for measuring the quality of the m-learning system in the present study, and these may impact students' perceived m-learning quality. Henceforth, the following hypothesized relationship can be derived.

**H9.** *"M-learning system quality" is a significant dimension which affects the students' "perceived m-learning quality".*

### 3.10. M-Learning Service Quality

Hassanzadeh et al. [74] considered service quality to be an essential component for the success of information systems and defined service quality as the user's requirements and how to meet these requirements effectively. Almaiah and Man [6] embraced the service quality as one of the factors used to investigate the quality of m-learning systems using key factors such as "trust, availability, responsiveness, and personalization", whereas the present study identified responsiveness, security, empathy, and usefulness as determining dimensions in students' perceived m-learning quality evaluation, and included the study of the impact on the quality of the m-learning service. In m-learning flexibility, transparency in services, clarity of instructions, cloud data storage availability, and being able to restart work from the point where, due to service interruptions, discontinuity occurred, are the factors incorporated in the evaluation of the quality of m-learning services; these may influence students' perceptions of m-learning quality. Henceforth, the following hypothesized relationship can be derived.

**H10.** *M-learning service quality as significant dimension affects the students' perceived m-learning quality.*

### 3.11. Perceived M-Learning Quality

MLQual model is conceptualized using 10 hypothesized relationships, and the final dimension of MLQual is perceived in terms of m-learning quality, which is evaluated through the 10 dimensions discussed in preceding paragraphs. The availability of effective m-learning system tools, trust in m-learning, achievement of educational goals, and a wish to continue m-learning in the future, as well as in hybrid modes of learning, are the factors identified for students' perceived quality of m-learning [26,67]. The proposed MLQual framework for m-learning quality evaluation focuses on two main aspects corresponding to MLSysQ and MLSerQ, which indicates that the evaluation of the quality of m-learning is carried out using the system and service quality assessment criteria.

## 4. Research Method

Research is a strenuous process of gathering and analyzing data in order to extract novel knowledge that can be used to endorse and extend existing theories [75]. Quantitative

research implicates the comprehensive review and evaluation of the existing literature for the formulation of hypotheses and the identification of appropriate constructs and variables [76]. Hence, the survey-based quantitative research approach was determined to be appropriate in the setting of the current study, which is positivist in nature [77].

*4.1. Participants and Survey Instrument*

An online survey-based questionnaire was publicized to the students studying in Indian universities. A Google form was designed for incorporating survey questions and survey link was created and shared among the students. Survey participants were undergraduate and postgraduate students in India who have been taking online sessions through mobile devices and hybrid modes of learning sessions. Participants' participation in the survey was anonymous and merely voluntary. The Google form for the survey response collection was circulated through a WhatsApp link to undergraduate and postgraduate students. The survey link was shared among those selected students who were taking the classes online. Academic colleagues played a key role in sharing the WhatsApp survey link with the students in their respective class sessions. Students were informed by the surveyor about the purpose and process of filling the survey. The survey was carried out from December 2021 to January 2022. Overall, 264 participants contributed to the survey and returned their replies. A sample size of 200 and above in quantitative exploration is considered adequate, specifically when SEM is being utilized [78]. The participant comprise (85%) participants falling under the age bracket of 20–23 years; however, the remaining (15%) participants fall under the age bracket of 24–25 years, where (65%) male and (35%) female students returned the survey responses. Students on engineering, computing, business administration, and science courses were targeted for the distribution of the survey. All these students have been taking online sessions for more than one and half years; hence, these have reasonable experiences to extend their perceptions about online sessions. We utilized a "purposive sampling strategy" since the students were conveniently reachable and had no personal prejudices [79].

A set of questionnaires as survey instrument was produced in this research to uncover the quality factors through students' perception to m-learning quality. Lee et al. [72] mentioned that Likert scales have been used in variety of research studies where students' questionnaire-based perceptions are sought. In order to assure the survey's applicability and proper context of the current research, questionnaires were generated from prior studies [22,26,73,79,80] associated with m-learning quality. The MLQual model consists of 11 constructs/dimensions and 48 measurement factors, and each construct has a number of related factors for measuring the constructs. The instrument's individual constructs with their identified associated factors are enumerated in Tables 1 and 2. A 5-point Likert scale is deliberated in the present study. Recorded responses from participants were categorized as "Strongly Disagree"; "Disagree"; "No Opinion/Neutral"; "Agree"; "Strongly Agree" [81].

*4.2. Pilot Study*

It was essential to carry out a pilot study to evaluate the consistency of the questionnaire. Thus, we examined the reliability of proposed scale through a pilot study which was conducted using randomly selected responses from 45 students. Cronbach's alpha test was employed to assess the internal consistency and reliability of the factors of the proposed constructs/dimensions. All the constructs were found to be reliable as their Cronbach's alpha values surpassed the prescribed acceptable range of 0.7; the acceptance criteria are classified in various categories (excellent if $\alpha_c \geq 0.9$, very good if $\alpha_c \geq 0.8$, good if $\alpha_c \geq 0.7$, adequate if $\alpha_c \geq 0.6$, poor if $\alpha_c \geq 0.5$, not acceptable if $\alpha_c < 0.5$) [82,83]. Since Cronbach's alpha test confirmed the reliability of the involved constructs, these became part of this concluded study. Table 1 shows Cronbach's alpha ($\alpha_c$) values of constructs used for the pilot study.

**Table 1.** Cronbach's alpha value of constructs.

| Constructs | Abbreviated Form Used | Cronbach's Alpha ($\alpha_c$) |
|---|---|---|
| Tangibility | Tan | 0.720 |
| Functionality | Fun | 0.730 |
| Reliability | Rel | 0.750 |
| Collaboration | Col | 0.800 |
| Responsiveness | Res | 0.700 |
| Security | Sec | 0.830 |
| Empathy | Emp | 0.760 |
| Usefulness | Use | 0.830 |
| M-learning System Quality | MLSysQ | 0.730 |
| M-learning Service Quality | MLSerQ | 0.790 |
| Perceived M-learning Quality | PMLQ | 0.880 |

*4.3. Structural Equation Modeling*

SEM is a significant statistical approach used for evaluating the hypotheses and relationships between specifically observed model constructs and latent variables [76]. AMOS 22 is used for implementing SEM on gathered data for examining the proposed hypotheses.

This research employed statistical methods for the measurement of instrument reliability and validity and then evaluated measurement model fit for the recommended MLQual conceptual research framework. The measurement model was evaluated by confirmatory factor analysis (CFA) to know whether the proposed constructs/dimensions were inclining towards reasonable validity, and whether consistency of the proposed scale is attained [84]. "Measurement model fit indices (GFI, AGFI, RMSEA, RMR, CFI, IFI, NFI, and TLI)" $\geq 0.85$ were considered for evaluating the measurement model fit [84,85]. Hypotheses testing was carried out in structural model fit measurement and hypothesized relationships amongst constructs of proposed model were determined by "path coefficient ($\beta$), critical ratio/*t*-value, and *p*-value" [76].

**5. Research Results**

Research results evaluation are broadly classified into two categories incorporating the "measurement model-fit and structural model-fit" assessments.

*5.1. Measurement Model Fit Evaluation*

Prior to examining the proposed theoretical framework and hypothesized relationships among the included constructs, the preliminary analysis was carried out via confirmatory factor analysis (CFA) and "reliability and validity" evaluations. CFA was implemented to determine the goodness-of-fit (GOF) of measurement model [76]. Table 2 shows the number of constructs and their identified factors used in this study. Three rounds of CFA were held with 11 constructs and identified 48 factors. After carefully examining the obtained results of each round, the items (Tan3, Res2, Sec3, Col2, Col3, and Emp3) were eliminated due to their low (<0.5) factor loadings [76]. Goodness-of-fit of measurement model was successfully achieved and confirmed the acceptance of 11 constructs and 42 items in the proposed framework. The final CFA round proven the statistically significant results with CMIN/DF (1.241), GFI (0.904), AGFI (0.889), TLI (0.971), CFI (0.976), IFI (0.978), NFI (0.888), RMR (0.05), and RMSEA (0.031). These results meet the measurement-model-fit criteria recommended by Kline [84]. Squared multiple correlation ($R^2$) values (0.404 to 0.726) were achieved within the acceptable range and standardized regression weights were obtained (>0.50), which further confirmed the measurement model fit [76].

The reliability of the anticipated measurement scale was established by means of Cronbach's ($\alpha$). Table 2 shows the estimated values of Cronbach's ($\alpha$) falling in between (0.718–0.827), which confirms the internal consistency of 42 factors in the proposed scale and fulfills Cronbach's ($\alpha \geq 0.7$) recommended criteria as suggested by Hair et al. [76]. In addition to reliability, the estimation of constructs validity was needed via achieving discriminant and convergent validity estimations. Standardized regression weight (factor loading) >0.50, average variance extracted (AVE) > 0.50, and composite reliability >0.70 are the key determinants for confirming the convergent validity of instruments [76,85]. All anticipated apparent items of the subsequent latent factors were found to be significant, with the composite reliability (CR) and AVE results surpassing the benchmarks correspondingly. Consequently, Table 2 illustrates the confirmed convergent validity.

**Table 2.** Convergent validity tests results.

| ITEMS | | Standardized Regression Weights | ($\alpha_c$) | CR | $R^2$ | AVE |
|---|---|---|---|---|---|---|
| Tan | Tan1 | 0.746 | | | 0.557 | |
| | Tan2 | 0.755 | 0.718 | 0.757 | 0.570 | 0.511 |
| | Tan4 | 0.636 | | | 0.404 | |
| Fun | Fuc1 | 0.693 | | | 0.480 | |
| | Fuc2 | 0.678 | | | 0.460 | |
| | Fuc3 | 0.752 | 0.805 | 0.842 | 0.566 | 0.516 |
| | Fuc4 | 0.719 | | | 0.517 | |
| | Fuc5 | 0.745 | | | 0.555 | |
| Rel | Rel1 | 0.744 | | | 0.554 | |
| | Rel2 | 0.697 | | | 0.486 | |
| | Rel3 | 0.731 | 0.749 | 0.814 | 0.534 | 0.523 |
| | Rel4 | 0.719 | | | 0.517 | |
| Col | Col1 | 0.755 | | | 0.570 | |
| | Col4 | 0.636 | 0.723 | 0.757 | 0.404 | 0.507 |
| Res | Res1 | 0.727 | | | 0.529 | |
| | Res3 | 0.697 | | | 0.486 | |
| | Res4 | 0.711 | 0.738 | 0.805 | 0.506 | 0.510 |
| | Res5 | 0.714 | | | 0.510 | |
| Emp | Emp1 | 0.711 | | | 0.506 | |
| | Emp2 | 0.693 | | | 0.480 | |
| | Emp4 | 0.729 | 0.770 | 0.808 | 0.531 | 0.513 |
| | Emp5 | 0.732 | | | 0.536 | |
| Sec | Sec1 | 0.723 | | | 0.723 | |
| | Sec2 | 0.721 | | | 0.520 | |
| | Sec4 | 0.698 | 0.789 | 0.810 | 0.487 | 0.516 |
| | Sec5 | 0.731 | | | 0.534 | |
| Use | Use1 | 0.745 | | | 0.555 | |
| | Use2 | 0.737 | | | 0.543 | |
| | Use3 | 0.740 | 0.771 | 0.819 | 0.548 | 0.531 |
| | Use4 | 0.691 | | | 0.477 | |

**Table 2.** *Cont.*

| ITEMS | | Standardized Regression Weights | ($\alpha_c$) | CR | $R^2$ | AVE |
|---|---|---|---|---|---|---|
| MLSysQ | MLSysQ1 | 0.674 | 0.766 | 0.819 | 0.454 | 0.532 |
| | MLSysQ2 | 0.695 | | | 0.483 | |
| | MLSysQ3 | 0.772 | | | 0.596 | |
| | MLSysQ4 | 0.771 | | | 0.594 | |
| MLSerQ | MLSerQ1 | 0.715 | 0.708 | 0.872 | 0.511 | 0.631 |
| | MLSerQ2 | 0.800 | | | 0.640 | |
| | MLSerQ3 | 0.853 | | | 0.728 | |
| | MLSerQ4 | 0.802 | | | 0.643 | |
| PMLQ | PMLQ1 | 0.755 | 0.827 | 0.858 | 0.570 | 0.600 |
| | PMLQ2 | 0.852 | | | 0.726 | |
| | PMLQ3 | 0.787 | | | 0.619 | |
| | PMLQ4 | 0.702 | | | 0.493 | |

The next step was to proceed for the determination of discriminant validity, and this further confirmed the measurement model fit. Table 3 clearly explains the attainment of discriminant validity as the diagonal values of Table 3 are higher than the residual column values.

**Table 3.** Discriminant validity results.

| | Tan | Fun | Rel | Col | Res | Sec | Emp | Use | MLSysQ | MLSerQ | PMLQ |
|---|---|---|---|---|---|---|---|---|---|---|---|
| Tan | 0.749 | | | | | | | | | | |
| Fun | 0.562 | 0.758 | | | | | | | | | |
| Rel | 0.493 | 0.516 | 0.763 | | | | | | | | |
| Col | 0.505 | 0.537 | 0.667 | 0.721 | | | | | | | |
| Res | 0.623 | 0.536 | 0.541 | 0.672 | 0.764 | | | | | | |
| Sec | 0.549 | 0.639 | 0.365 | 0.535 | 0.585 | 0.724 | | | | | |
| Emp | 0.661 | 0.609 | 0.596 | 0.560 | 0.667 | 0.670 | 0.821 | | | | |
| Use | 0.493 | 0.517 | 0.543 | 0.601 | 0.596 | 0.530 | 0.581 | 0.728 | | | |
| MLSysQ | 0.612 | 0.636 | 0.528 | 0.632 | 0.563 | 0.493 | 0.577 | 0.589 | 0.709 | | |
| MLSerQ | 0.589 | 0.536 | 0.571 | 0.592 | 0.621 | 0.639 | 0.598 | 0.613 | 0.636 | 0.802 | |
| PMLQ | 0.520 | 0.588 | 0.648 | 0.621 | 0.534 | 0.590 | 0.672 | 0.515 | 0.627 | 0.669 | 0.815 |

### 5.2. Structural Model Fit Evaluation

After the successful execution of the measurement model, the next step was to evaluate the structural model to confirm the hypothesized relationships among the constructs of the proposed framework. Path analysis by utilizing AMOS was carried out to confirm the proposed hypothesized relationships. Figures 2 and 3 demonstrate the results of 11 hypothesized relationships amongst the suggested constructs and illustrate the fitness of the structural model.

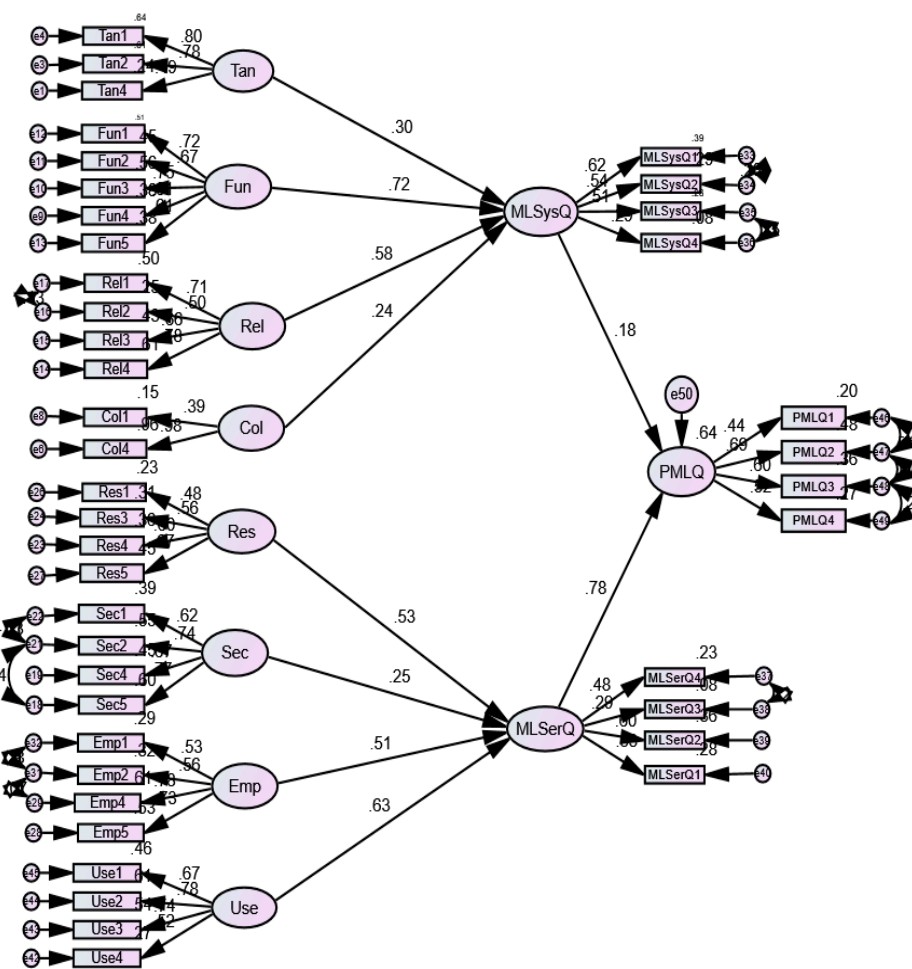

**Figure 2.** Path analysis using structural model fit evaluation.

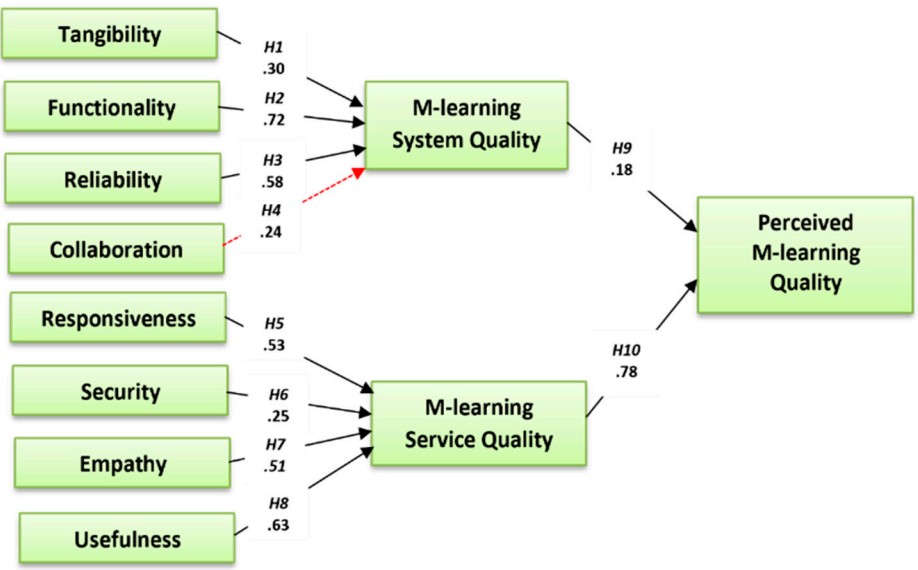

**Figure 3.** Validated M-learning quality (MLQual) framework.

Table 4 shows the results hypotheses testing acquired from structural model fit evaluation. These contained computations of "*t*-values or critical ratio (C.R. > 1.96), path coefficient (β), and *p*-values" at (* $p \leq 0.05$; ** $p \leq 0.01$; *** $p \leq 0.001$) given by Hair

et al. [76]. Overall, 1 path coefficient value was found inconsequential out of 10 path coefficient values, and consequently it was dropped from the proposed framework.

**Table 4.** Hypothesis testing results.

| Path | CR/ $t$-Value | Path Coefficient ($\beta$) | $p$-Value | Outcome of Hypothesis | |
|:---:|:---:|:---:|:---:|:---:|:---:|
| Tan → MSysQ | 3.740 | 0.30 | *** | H1 | Accepted |
| Fun → MSysQ | 7.091 | 0.72 | *** | H2 | Accepted |
| Rel → MSysQ | 6.892 | 0.58 | *** | H3 | Accepted |
| Col → MSysQ | 1.526 | 0.24 | 0.127 | H4 | Rejected |
| Res → MSerQ | 5.471 | 0.53 | *** | H5 | Accepted |
| Sec → MSerQ | 3.716 | 0.25 | *** | H6 | Accepted |
| Emp → MSerQ | 6.017 | 0.51 | *** | H7 | Accepted |
| Use → MSerQ | 6.762 | 0.63 | *** | H8 | Accepted |
| MSysQ → PMLQ | 2.562 | 0.18 | 0.010 | H9 | Accepted |
| MSerQ → PMLQ | 4.951 | 0.78 | *** | H10 | Accepted |

After achieving positive execution of the measurement model and the structural model, we found below the validated form of the proposed hypothetical framework which we proposed for assessing the m-learning quality. Figure 3 shows the established framework, which illustrates the positive relationships among various constructs. However, the relationship between collaboration (Col) and m-learning system quality (MLSysQ) was not determined as it failed to meet the acceptance criteria.

## 6. Discussion

This contemporary pragmatic study emphasizes the elements associated with the evaluation of m-learning quality during the COVID-19 pandemic from the students' perspective. Furthermore, this study is going to be effective even after the COVID-19 pandemic for assessing the m-learning quality from the students' perspective. After a comprehensive study of the literature, we embraced 11 constructs with 42 items in our framework. Analysis determined the influence of 10 constructs including (tangibility, functionality, reliability, collaboration, responsiveness, empathy, security, usefulness, m-learning system quality, and m-learning service quality) on students' perceived m-learning quality. Hypothesized relationships among the constructs were identified via path coefficient ($\beta$) values occurring in Figures 2 and 3. Moreover, Table 4 clearly demonstrates the hypothesized examination findings. The findings of this empirical study show that the proposed MLQual framework may accurately reflect the collected data and aid in understanding students' perceptions of the quality of their m-learning system. The relevance of each construct of the MLQual is then addressed, as determined by analysis of the hypotheses. Table 4 illustrates the results of hypothesis testing.

Table 4 indicates that all hypotheses from H1 to H10 were established, excluding H4 which was rejected due to its inappropriate $p$-value (.127). H4 did not meet the specified $p$-value criteria (* $p < 0.05$, ** $p < 0.01$, *** $p < 0.001$), implying that the "collaboration" (Col) did not certainly impact the m-learning system quality. While running the measurement model fit it was observed that Col2 and Col3 were dropped due to low factor loadings. Rejection of H4 hypothesis indicates that the construct collaboration (Col) did not influence the m-learning system quality (MSysQ). Therefore, dismissal of hypothesis H4 implies that m-learning environment did not support the collaborative learning during COVID-19 within the context of m-learning in India. Hence, maintaining the quality of m-learning needs an improved collaborative learning environment so that students and instructors could work in a group for performing group/team activities [86].

Path analysis among (Tan → MLSysQ) shows the decisive effect of tangibility on MLSysQ as hypothesized in H1 and this relationship established with ($\beta$ = 0.30, t = 3.740, *** *p*) values. This relationship is confirmed in the study performed by Akhlaghi et al. [64]. where tangibility was used as a construct and assessed using factors such as physical facilities, equipment, personnel, and communication materials. Accessibility to m-learning applications with appropriate tangible items such as laptop, desktop, mobile devices, sufficient memory size, and fast and continuous internet offer greater interactivity with the m-learning system and enhance the m-learning system quality, which further influences the students' learning [87].

Hypothesized relationship H2 among (Fun → MLSysQ) indicates the positive effect of functionality on MLSysQ with accepted values ($\beta$ = 0.72, t = 7.091, *** *p*) and this relationship is confirmed via previous studies [6,11,65,66] carried out to exhibit the role functionality and its impact on m-learning process while students use the m-learning systems. Universities extended the use of m-learning systems during the COVID-19 pandemic, and these are Web-based systems which can be accessed through Web-based applications [26]. Therefore, the functionality of the Web-based system including required functions, interactivity, fast navigation between Web pages, a user-friendly environment, and flexibility in using the m-learning systems are considered to be vital factors constituting quality in the m-learning system [2]. These functionality items are sought by students while they engage in m-learning; therefore, these factors are included in the present study.

Path analysis concerning (Rel → MLSysQ) implies the positive hypothesized relationship H3 between the reliability of the m-learning system and the quality of the m-learning systems with received values ($\beta$ = 0.58, t = 6.892, *** *p*). This indicates that the reliability of mobile devices, the quick upload and download of study material, and accurately performing the transaction irrespective of students' locations are important criteria which comprise the reliability of m-learning and influences the quality of the m-learning system [67].

The hypothesized relationship (H5) among (Res → MLSerQ) reveals the positive influence of responsiveness on m-learning service quality with received values ($\beta$ = 0.53, t = 5.471, *** *p*). This hypothesized relationship is confirmed by previous findings [6,26]. Responsiveness in this study is related to the response to students in terms of quick IT support, availability of easy and clear online study material, and provision of instant information availability to the students, quick response to students' group work, and responses related to changes in schedule. These factors make up responsiveness and influence the quality of m-learning services also mentioned in previous studies [70,88].

Path analysis between (Sec → MLSerQ) confirmed the hypothesized relationship (H6) between the security of m-learning system and m-learning service quality using accepted values ($\beta$ = 0.25, t = 3.716, *** *p*). Security constructs in this study cover factors such as confidentiality of students' personal information, transaction safety, frequent security updates, no misuse of personal information, and sufficient confidence of students. One study by Sletten and Montebello [89] entitled "secure m-learning" highlighted these measurement factors and their influence towards m-learning.

Path analysis among (Emp → MLSerQ) shows the positive hypothesized relationship H7 between empathy and m-learning service quality with established values ($\beta$ = 0.51, t = 6.017, *** *p*). The relationship of empathy between m-learning and m-learning quality was also explained by Stodnick and Rogers [70] and Isaias, Reis, Coutinho and Lencastre [88] in their studies. In the present study, item empathy in m-learning focused on attention to each student, encouragement of students, students' needs, and proper advising by the instructor during their office hours.

The hypothesized relationship (H8) among (Use → MLSerQ) indicates the positive influence of the usefulness of the m-learning system on the m-learning service quality with established values ($\beta$ = 0.63, t = 6.762, *** *p*). Previous studies [71,73] also confirmed such a relationship. The usefulness of the m-learning system in the context of the present study embraced factors such as quick and easy interaction, time management, tracking the weekly tasks, and helping to recover the students' study gap. The presence of these

factors in the assessment of their usefulness shows the significant influence of usefulness on m-learning service quality.

Path analysis amongst (MLSysQ → PMLQ) implies the positive hypothesized relationship (H9) between the quality of the m-learning system and perceived m-learning quality with obtained values (β = 0.18, t = 2.562, ** *p* value ≤ 0.010). The relationship between these was aligned with similar studies carried out by various authors [6,26,65]. The M-learning system in the present study examined factors including mobile devices which are compatible with the operating system, application technicality, screen resolution, and speed to process heavy transactions. These factors constitute the quality of the m-learning system and influence on the overall students' m-learning quality.

The hypothesized relationship H10 amongst (MLSerQ → PMLQ) implies the clear effect of m-learning service quality on students' perceived m-learning experience with the obtained values (β = 0.78, t = 4.951, *** *p*). Similar studies by [6,26] confirmed the relationship of service quality with m-learning quality. The use of mobile devices is found to be flexible for online classes. Transparent services, clear instructions, cloud data storages, and offering uninterrupted services to measure the quality of the m-learning service were found to be important factors in m-learning.

Based on preceding analysis on the hypothesized relationship among various quality constructs, the following directions are given for delivering a better h-learning experience to students and enhance students' engagement.

- Develop a reliable m-learning platform that can deliver educational materials and services to the students without time and place constraints. Additionally, the m-learning system can make sufficient coordination among instructors, students present in classroom, and students attending classes virtually, which means in hybrid modes of learning (h-learning).
- Improve functionality of the m-learning system so that it could give the students a better interactive experience while they take their courses in h-learning mode. The M-learning system should include all the possible features such as interactivity, user-friendly environment, reliable system, uninterrupted connection, and quick navigation among software screens and files.
- Academics should give more emphasis to the "collaboration" factor while designing the m-learning/h-learning system in order to develop a significant collaborative environment to be offered for the students to carry out their group work activity such as group assignments, projects, and group discussions using live audio or video chat.
- The M-learning system should provide students with quick accessibility to updated digital resources such as e-books, videos, discussion forums, and other online materials.
- As a part of student-centered m-learning, the feedback mechanism should be designed in a way that students will be able to receive feedback on their queries, quizzes, assignment, and projects in prompt manner from their instructors. Furthermore, instructors should be available online via the m-learning system during posted office hours to solve the students' issues. This will boost students' confidence in m-learning/h-learning.
- Students need continuous attention in the m-learning process; therefore, serious consideration should be given to feature "responsiveness" in m-learning systems. Here, continuous IT support plays a significant role in raising students' motivation towards engaging in the m-learning/h-learning process.
- Students' personal information should be handled carefully and kept confidential in the m-learning process. Therefore, privacy and security features should be essential parts of the m-learning system.

Comparing the validated MLQaul results with the previous studies [45,48,52,58–60,63] in the literature, it is confirmed that the number of constructs which were identified in MLQual, show positive correlations and our research results are consonance to the previous findings. As SERVQUAL [63] is considered as the base model in our study so its extended version MLQual confirms the validity of quality constructs "reliability, responsiveness,

tangibility, assurance, and empathy" within the m-learning domain. Validated constructs of MLQual including tangibility [64], functionality [11,26,65], reliability [67,68], collaboration [22], security [58], empathy [63,69,70], usefulness [71–73], m-learning quality [6,74], m-learning service quality [7,65], and perceived m-learning quality [26,67] are found to be aligned with previous points of research. Analysis has revealed further information on construct "collaboration" which did not show positive impact on m-learning quality. This was due to the lack of collaboration students experienced among their peers in the online learning environment as mentioned in [86]. The MLQual framework presents a novel approach and criteria for evaluating the quality of m-learning, and compares values with the existing m-learning literature. Additionally, the MLQual framework gives significant directions to improve the h-learning quality.

## 7. Theoretical Implications

The SERVQUAL model entitled "multiple items scale for measuring consumer perceptions of service quality" is presented in [63]. This model has been employed widely for measuring consumer perception towards offered service quality. However, its implementation is inadequate in evaluating m-learning quality from students' perspectives in the higher education sector. Our present research extended the SERVQUAL model and validated MLQual in an m-learning context. Considering the theoretical implication of our study, MLQual incorporates improved quality evaluation criteria and evaluates the m-learning quality from students' perspectives in a comprehensive manner. Furthermore, considering the advanced technological learning environment, MLQual provides novel quality constructs such as "functionality, collaboration, security, usefulness, m-learning system quality, m-learning service quality, and perceived m-learning quality". Through extending SERVQUAL philosophy, current research work investigated the unique elements that influence m-learning quality. All quality-related factors that ought to be a part of the m-learning process in higher education are covered by MLQual. Additionally, this study expedites and lays the groundwork for subsequent research that might be focused on:

- Determination of effective teaching methods in (m-)/(h)-learning.
- Reproduction of this work in various perspectives in order to evaluate the findings.
- Implementation of the study's key findings in order to develop teacher training programs for h-learning.

## 8. Practical Implications

Higher education institutions may take benefit of this proposed MLQual framework to lead their future m-learning policies and strategies. Furthermore, universities may assess their current efforts in providing online/hybrid modes of educational instruction. The university administration can invest in the necessary upgradation of essential technical infrastructure. This will ensure the successful implementation of m-learning technologies in offered academic programs. Likewise, this will also ensure that students will receive the utmost quality of m-learning services throughout the academic year. Considering the m-learning quality criteria suggested in this study, instructors may design course materials by maintaining the m-learning quality and expand students' engagement in m-learning.

## 9. Limitations and Future Research Directions

The COVID-19 pandemic forced institutions to take a shift towards m-learning from traditional learning, particularly in the setting of Indian universities [26]. M-learning was fully implemented at a particular time during the COVID-19 pandemic. H-learning started later on, around the same time as the students' gradual return towards universities campuses. Therefore, we need to further evaluate students' perceptions towards hybrid modes of learning. In spite of key contributions, this study has two more limitations. The first is the homomorphism of the data, and the second is the sample size which is restricted to a specific socio-demographic area within India. Therefore, MLQual could be validated by considering higher education institutions of other Indian territories. Furthermore, the

inclusion of instructors' viewpoints will play a vital role in determining factors while evaluating m-learning quality in hybrid modes of learning.

Considering the previously discussed limitations of the present study, this study requires future research work to overcome these limitations. Students' behavior and perception towards m-learning quality varies within the country as well as in different countries too, which is possibly due to cultural and socio-technical differences. Consequently, for the future research work, there is a need to further validate the proposed MLQual framework by carrying out studies in other Indian states or in other countries' h-learning settings. Future research work could also include revalidation of the MLQual framework by performing comparative studies of two different regions within the country or exclusively by considering two different countries.

## 10. Conclusions

An unprecedented pandemic situation turned conventional classroom teaching into online teaching. This shift raised a need among academic institutions to deliver the quality of an m-learning academic environment. Hence, it was important for the academic institutions to realize what students perceive with regards to m-learning quality in hybrid modes of learning. Two research questions were identified: the first one was "what are the factors comprising the students' perception towards m-learning quality within the context of higher education", and the second was "are there significant associations between the m-learning quality factors and students' utilization of m-learning system". This intensive research identified and examined the factors that are considered for evaluation of students' perceived quality of m-learning, and this empirical study also developed the MLQual framework. MLQual offers an expansion to the SERVQUAL model and incorporated constructs (tangibility, functionality, reliability, collaboration, responsiveness, empathy, security, usefulness, MLSysQ, MLSysQ, perceived m-learning quality). Measurement model fit and structural model fit indices confirm the associations among the proposed constructs in MLQual framework. The research outcomes further support the enhanced SERVQUAL model's applicability in the field of m-learning. Today's technological environment offers a variety of ways to design online contents for m-learning. Therefore, it is essential to understand students' perceptions and their preferences for making the m-learning and delivery process productive and efficient. The suggested framework, MLQual, aims to broaden the established theories in the area of m-learning quality assessment.

**Author Contributions:** Conceptualization, S.F.H.Z. and A.K.; Methodology, A.K. and S.F.H.Z.; Validation, S.F.H.Z. and A.K.; Formal analysis, S.F.H.Z.; Investigation, S.F.H.Z. and V.O.; Data curation, S.A.H.Z.; Writing—original draft, V.O. and S.A.H.Z.; Writing—review & editing, V.O. and S.A.H.Z.; Visualization, A.K. and S.A.H.Z.; Project administration, S.F.H.Z. All authors have read and agreed to the published version of the manuscript.

**Funding:** This research received no external funding.

**Institutional Review Board Statement:** Not applicable.

**Informed Consent Statement:** Informed consent was obtained from all subjects involved in the study.

**Data Availability Statement:** Data will be available upon request from the corresponding author.

**Conflicts of Interest:** The authors declare no conflict of interest.

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
