# Peer review of "Students’ Perceived M-Learning Quality: An Evaluation and Directions to Improve the Quality for H-Learning"

_education, doi:10.3390/educsci13060578_

Round 1

Reviewer 1 Report

Dear Authors,

I received for the review process your article "Students’ perceived m-learning quality: An evaluation and directions to improve the quality for h-learning" from the Education Sciences Journal.

The Introduction section is very dense and well written, but it seems to omit exactly the most important aspects related to such a chapter. Thus, I recommend you to add new paragraphs to describe the research gap, the research goal and the research questions. By presenting these aspects, the reader will understand the focus of your article. I also recommend you to insert a distinct paragraph at the end of the Introduction so that you describe the general sections of your article.

Within section "2.1. Learning techniques in higher education" you should add many relevant references. I recommend you to include the following resources: https://doi.org/10.1007/s10758-019-09408-7, https://doi.org/10.24818/ie2020.02.01, https://doi.org/10.1007/s10639-022-10925-9, https://doi.org/10.1007/s10734-021-00748-x.

I think that "Figure 1. M-learning quality (MLQual) evaluation framework" should be placed after the descriptions of all the research hypotheses. Please think about the opportunity of moving the figure after the line 359 in your manuscript.

In the section "4.1. Participants and survey instrument", at the lines 373 - 375 you say that "Participants’ participation in the survey was anonymous and merely voluntary. 264 participants contributed to the survey and returned their replies." Please provide many details: how did you select the participants? When did you conducted the surveys?

The title of "Table 3. Discremenant validity results" contains a typo-mistake. Please revise the word "discremenant".

I recommend you to update the title of the section "9. Limitations" to "9. Limitations and Future Research Directions". Here you should also present the future research directions. Please think about longitudinal studies, comparative studies between regions or countries or types of populations.

Dear Authors, I think that if you address all the above recommendations, the article will have an increased quality.

Best regards!

Reviewer 2 Report

The chapter presents a relevant and innovative topic. The research presents a clear and objective methodology. The research did not present the problem and its objective. I suggest that it be inserted in the abstract. However, I suggest that a section on h-learning be inserted, as well as the final considerations in the article. 

Round 2

Reviewer 1 Report

Dear Authors,

The revised  article was improved and I appreciate your work. After reading the current version of the manuscript, I have some minor recommendations:

- the sentence from the page 4, lines 197 - 198 "As previously exclaimed, h-learning integrates both face-to-face and online learning." should be corrected. Please use "explained" instead of "exclaimed". It seems to be a typo-mistake here.

- in table 2 at pages 12 - 13 you present the AVE values. It is not mandatory, but I suggest you to represent these values as absolute values (0.511) instead of percentages (51.5%). Usually, these values are presented in the absolute manner in the modern scientific articles.

- in the section "6. Discussion", I recommend you to also include a short discussion about a comparison between your research results and the previous results from the literature.

Best regards!

-
